# Toward dimensional psychiatry in youth: A data-driven analysis of transdiagnostic internalizing symptoms in childhood and adolescence

Ines M. Engler[1,2]*, Nicolas Langer[1,2]

**1** Methods of Plasticity Research, Department of Psychology, University of Zurich, Zurich, Switzerland,
**2** Neuroscience Center Zurich (ZNZ), University of Zurich and ETH Zurich, Zurich, Switzerland

* ines.engler@uzh.ch

## Abstract

Categorical diagnostic systems for psychopathology, such as the DSM and ICD, have long been criticized for their limited validity and reliability. Dimensional models, like the Hierarchical Taxonomy of Psychopathology (HiTOP), offer an alternative by focusing on transdiagnostic dimensions that better capture the complexity of mental health disorders. While HiTOP's internalizing spectrum has been studied extensively in adults, its applicability and structure in children and adolescents remain less clear. Further, understanding sociopsychological indicators associated with internalizing dimensions in this age group could improve developmental psychopathological interventions. We analyzed data from 4,142 participants aged 5–21 (65.7% male; mean age = 10.46) from the Healthy Brain Network. Using exploratory and confirmatory factor analyses, we tested the internalizing structure proposed by HiTOP, with an additional focus on invariance across sex, age, and diagnostic groups. The hierarchical structure was tested through hierarchical CFA and the extended Bass-Ackward method. Structural equation modeling (SEM) examined latent factor relationships, and sociopsychological variables associated with the factors. A four-factor structure was identified: *Distress*, *Nervousness*, *Social Fears*, and *Obsessions and Compulsions* (*OC*). The model demonstrated partial invariance and strong fit indices. Sociopsychological variables, i.e., predictors and a quality of life indicator of the factors, included parental attitudes, discipline, bullying, and daily functioning. DSM categories and CBCL scores mapped well onto the latent factors. These findings suggest the potential clinical utility of a dimensional model for internalizing disorders in youth. Future studies should further examine the role of sociodemographic factors on dimensional constructs, explore predictive developmental trajectories longitudinally, and verify the structure of all HiTOP spectra across age groups to advance dimensional models in pediatric psychopathology research, as well as their implementation in clinical practice.

**Data availability statement:** The data and code utilized for all analyses can be found at https://osf.io/br8wd/files/osfstorage.

**Funding:** This work was supported by the Swiss National Science Foundation (SNSF) Grant No. 10001C_220048, awarded to NL (https://www.snf.ch/en). This grant includes the salary of multiple collaborators, including IE. The funders had no role in study design, data collection and analysis, decision to publish or preparation of the manuscript.

**Competing interests:** The authors have declared that no competing interests exist.

# 1. Introduction

The prevalence of mental health disorders is increasing at a concerning rate globally, reaching unprecedented levels and causing a significant social, physical health, and economic burden [1,2]. A significant challenge in current mental health research is the reliance on group-average case-control studies and traditional diagnostic frameworks as those exemplified by classification systems of psychopathology [3,4]. Systems such as the International Classification of Diseases (ICD) and the Diagnostic and Statistical Manual of Mental Disorders (DSM) define, structure, and categorize mental health disorders based on symptom-based diagnostic criteria [5,6]. These frameworks facilitate communication among clinicians, patients, and researchers, while supporting the diagnostic processes and treatment selection [7,8]. However, numerous limitations of conventional nosologies have been highlighted, including the high comorbidity between many mental disorders [9,10], significant heterogeneity within syndromes [11–13], and the low to moderate reliability of diagnoses, such as low inter-rater and test-retest reliability [4,14,15]. Furthermore, many undiagnosed or subclinical cases remain unaddressed due to the polythetic and dichotomous nature of these categorical classification systems [16,17]. Importantly, emerging research from genetics, systems neuroscience, and behavioral science has increasingly challenged the validity of traditional diagnostic labels [18–20].

In response to these issues, alternative nosological systems have emerged, e.g., clinical staging models [21], network models [22], and the Research Domain of Criteria (RDoC) [23], among others. One prominent example is the Hierarchical Taxonomy of Psychopathology (HiTOP) Consortium's project, which aims to overcome the shortcomings of traditional classification systems by introducing a hierarchical, dimensional, and empirically driven model that maps a wide range of interconnected psychopathologies [3]. HiTOP organizes psychopathological traits and symptoms into hierarchically ordered transdiagnostic dimensions. At the base of the hierarchy lie lower-order traits and symptoms, which are grouped by their co-occurrence into syndromes. These syndromes are further clustered into subfactors, which are then nested within broader transdiagnostic spectra. Five spectra have been differentiated: *somatoform*, *internalizing*, *thought disorders*, *detachment*, and *externalizing*. At the top of the model, a general factor of psychopathology, the *p-factor*, presides over all other dimensions, which "represents features common across all of psychopathology", as well as a vulnerability underlying all psychopathologies [3,24]. However, the existence and validity of the p-factor have been questioned, as they depend on the chosen statistical method and the structural conceptualization of a dimensional model [25,26]. Nonetheless, this hierarchical and dimensional structure allows HiTOP to capture the complexity and interconnectivity of mental health conditions more accurately than traditional diagnostic systems [27,28].

In the HiTOP model, the conceptual basis of the dimensions is a separation between inherently internalizing and inherently externalizing disorders. This principle follows a long-standing finding from childhood studies by Thomas Achenbach, whereby "externalizing" refers to outwardly expressed symptoms, and "internalizing" to inwardly directed ones [29]. Likewise, HiTOP's externalizing spectrum represents

a range of outward maladaptive behaviors, whereas the internalizing spectrum represents an inward manifestation of emotional dysfunction [30]. It encompasses four subfactors: *Distress*, *Fear*, *Sexual Problems*, and *Eating Pathologies*. The internalizing domain is particularly important due to its wide scope, its association to impaired functioning (e.g., executive functioning or disability level), the high degree of comorbidity among its related disorders, and its raising prevalence [31–33].

Albeit the basic dichotomy between an internalizing and an externalizing spectrum initially emerged from childhood studies, most research on HiTOP has concentrated on the structure of psychopathology in adulthood. As a result, the applicability of HiTOP to early childhood and adolescence remains largely unexamined, despite these being critical developmental periods marked by substantial neural reorganization, and during which many mental health disorders first emerge [34,35]. A recent exploratory study has begun to address this gap by investigating the validity of the broader HiTOP structure in healthy children and adolescents, as well as in a clinical referral sample. Here, only a small proportion of their sample was younger than 11 years of age, an age-group that is often under-examined in internalizing investigations, and the model was tested for the complete group, without examining model aptitude for subgroups (e.g., sex, age) within the sample. The authors identified four domains that are closely related to HiTOP but did not fully replicate its spectral structure [36]. Consequently, the robustness of detailed symptom-level models within clinical pediatric populations and across different measurement methods remains mostly uncertain.

The present study aims to address current shortcomings and challenges in the field of youth mental health research through the analysis of a large clinical pediatric sample (N = 4,142). With this purpose, the study investigates internalizing symptoms and traits to identify and validate HiTOP's transdiagnostic structure of internalizing psychopathology, while also evaluating the model's suitability across the full age range from childhood to late adolescence. Additionally, we investigate the impact of internalizing symptoms on daily functioning, as defined by the WHO, and identify potential predictive risk factors. Overall, this study aims to inform the adaptation of the HiTOP model for younger populations with internalizing mental health problems and establish its viability for future clinical implementation. This may ultimately contribute to more targeted mental health interventions and improved outcomes in younger generations.

## 2. Methods

### Ethics statement

The analyses included data from 4,830 participants from the Healthy Brain Network (HBN) [37]. The HBN is an ongoing large-scale multimodal pediatric data acquisition project conducted by the Child Mind Institute in the New York area (USA). The Healthy Brain Network (HBN) study was approved by the Chesapeake Institutional Review Board (https://www.chesapeakeirb.com/), and the approval explicitly includes permission for secondary data analyses. Data access requires completion of the HBN Data Usage Agreement (DUA), which we obtained prior to conducting the present analyses. Prior to conducting the research, written informed consent was obtained from participants ages 18 or older. For participants younger than 18, written consent was obtained from their legal guardians and written assent obtained from the participant. The current study involved secondary analysis of de-identified data, accessed on February 26, 2024, and did not involve direct interaction with human participants.

### Sample

Sample characteristics, such as age, sex, race, number of diagnoses and CBCL T-scores, are displayed in Table 1.

### Measures

The 6,126 available behavioral and demographic variables of the HBN were first screened for thematic and analytical aptitude: variables on physical, bodily and cognitive measures, protocol items, age-specific questionnaires, demographic information (including basic demographics and risk factors), and non-internalizing symptoms and traits, according to their

**Table 1. Sample Characteristics.** Shown are number of observations, age range and mean age, the total and percentual number of male observations, the total and percentual number of different races and ethnicities, and the total and percentual number of diagnoses and diagnostic subgroups. "Clinical Diagnoses" are DSM-based categorical diagnoses, whereas "Clinical Scores" are CBCL-based T-scores. Note that "Externalizing" in the DSM refers to neurocognitive and neurodevelopmental disorders, i.e., ADHD or ASD, in addition to behavioral disorders, whereas the CBCL Externalizing score only includes behavioral, aggressive and disruptive disorders. For comparability, the CBCL's Attention Problems T-Score is also included, which represents neurodevelopmental disorders.

| Measure | HBN Sample | DS1 | DS2 |
|---|---|---|---|
| N | 4,830 | 2,045 | 2,097 |
| **Age** | | | |
| Range | 5.00 to 21.90 | 5.06 to 21.90 | 5.08 to 21.51 |
| Mean | 10.31 | 10.38 | 10.35 |
| **Sex** | | | |
| N (%) | ♂3,071 (63.6%) | ♂1,302 (63.7%) | ♂1,333 (63.6%) |
| **Race and Ethnicity** | | | |
| White/ Caucasian | 2,276 (47.12%) | 1,089 (53.25%) | 1,060 (50.55%) |
| Black/ African American | 587 (12.15%) | 260 (12.71%) | 274 (13.07%) |
| Hispanic | 401 (8.30%) | 185 (9.05%) | 180 (8.58%) |
| Asian | 132 (2.73%) | 58 (2.84%) | 62 (2.96%) |
| Indian | 28 (0.58%) | 8 (0.39%) | 16 (0.76%) |
| Two or more races | 711 (14.72%) | 319 (15.6%) | 350 (16.69%) |
| Unknown or otherwise specified | 695 (14.39%) | 126 (6.16%) | 155 (7.39%) |
| **Clinical Diagnoses (DSM)** | | | |
| N (%) | 3,845 (79.61%) | 1,772 (86.65%) | 1,816 (86.60%) |
| Internalizing (%) | 896 (18.55%) | 407 (19.90%) | 432 (20.60%) |
| Externalizing (%) | 2,949 (61.06%) | 1,365 (66.75%) | 1,384 (66%) |
| No Clinical Diagnosis (%) | 390 (8.07%) | 168 (8.22%) | 168 (8.01%) |
| Incomplete Evaluation (%) | 595 (12.32%) | 105 (5.13%) | 113 (5.39%) |
| **Clinical Scores (CBCL)** | | | |
| Mean Int. T-Score (SD) | 56.60 (11.31) | 56.79 (11.41) | 56.40 (11.24) |
| Mean Ext. T-Score (SD) | 54.97 (11.75) | 55.00 (11.83) | 54.87 (11.63) |
| Mean AP T-Score (SD) | 62.90 (9.77) | 62.94 (9.69) | 62.82 (9.83) |

DSM = Diagnostic and Statistical Manual of Diseases; DS1 = Dataset 1; DS2 = Dataset 2; Int = Internalizing; Ext = Externalizing; AP = Attention Problems.

source questionnaires, were first excluded. After, variables with more than 30% of missing data, retrospective and self-report variables were discarded, in addition to variables with too little variance (e.g., high single-category endorsement, i.e., variables that had 85% of answers in the same answer category) and that have similar wording (e.g., variables "Cries often" and "Often cries") to reduce redundancy. Additionally, variables with low sampling adequacy (i.e., their Kaiser-Meyer-Olkin value was < 0.80) [38], and with factor loadings ≤0.20 were removed (see Fig 1). This selection resulted in 78 parent-reported variables with acceptable distributions and missingness (≤22% per variable, mean missingness ≈11.6%) and thematic coherence to the subfactors of the internalizing spectrum as presented in the HiTOP model. The remaining missing data, which was not fully missing at random (i.e., some participants did not receive certain questionnaires if the study clinicians did not deem them necessary) was imputed as described below (see Data Preprocessing). The 78 variables stemmed from six different questionnaires (Child Behavior Checklist, Extended Strengths and Weaknesses Assessment of Normal Behaviors, Mood and Feelings Questionnaire, Screen for Anxiety Related Disorders, Strengths and Difficulties Questionnaire, HBN's Symptom Checker; see Table 2) were selected for factor analysis (FA), as well as nine additional variables representing possible sociopsychological predictors of internalizing psychopathology for structural

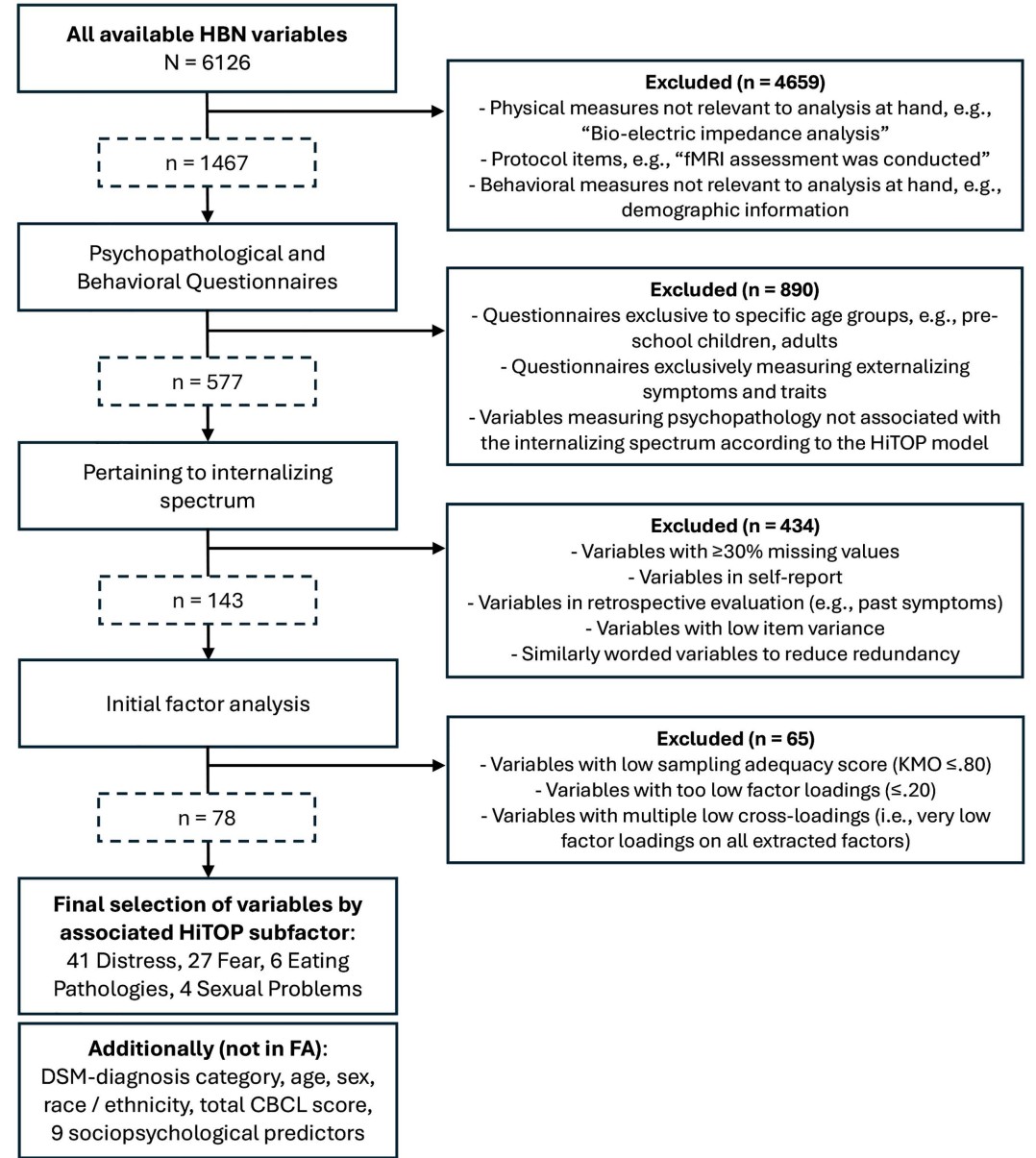

**Fig 1. Variable Selection Procedure.** All available HBN variables were screened following data- and content-driven approaches. Variables were sequentially excluded based on different criteria, following broader (e.g., "non-internalizing variables") to more precise (e.g., "KMO ≤ 0.8") exclusion criteria. The final selection included 78 variables, each one assigned to the four HiTOP subfactors, as well as an additional set of predefined demographic and clinical variables utilized in further analyses, but not in the factor analysis. FA = Factor Analysis.

equation modeling (SEM). A sum score variable, reflecting daily functioning, was selected as an indicator variable for SEM from the WHO Disability Assessment Schedule (WHODAS), which we refer to as a "quality of life (QoL) indicator" [39]. The variable contents, scale and source questionnaires are listed in Table 2. For further analyses beyond FA and SEM, the youth's age, sex, race, their pubertal timing, and the diagnostic categories of their DSM-based diagnoses, as well as the Child Behavior Checklist's (CBCL) sum scores were considered.

**Table 2. Study Variables. Variable IDs, Contents, Scale and Source Questionnaires of all 78 variables utilized in EFA, CFA and MGA. For copyright reasons, the full variable as it appears in some questionnaires when presented to participants cannot be reprinted here. Thus, a shortened version of every variable, or the construct the variable represents, can be found in the second column. The third column shows the ordinal Likert scale of each variable. The fourth column indicates the original publications on each questionnaire, and the source of the description used in the second column. The first column lists variable ID for reproducibility purposes.**

| Variable | Construct/ Content | Scale | Questionnaire and Literature |
|---|---|---|---|
| CBCL_05 | Enjoys little | 0–2 | **Child Behavior Checklist (**S1 Checklist**)** |
| CBCL_09 | Obsessions | 0–2 | Achenbach & Edelbrock, 1983; Dedrick et al., 1997; |
| CBCL_12 | Complains of loneliness | 0–2 | Gomez & Vance, 2014; |
| CBCL_14 | Crying | 0–2 | Ivanova et al., 2007; |
| CBCL_24 | Not eating well | 0–2 | Nakamura et al., 2009 |
| CBCL_25 | Animosity | 0–2 | https://www.apa.org/depression-guideline/child-behavior-checklist.pdf |
| CBCL_29 | Many fears | 0–2 | https://aseba.org/aseba-subsequent-developments/ |
| CBCL_33 | Complains about being unloved | 0–2 | |
| CBCL_35 | Feels worthless | 0–2 | |
| CBCL_42 | Is rather alone | 0–2 | |
| CBCL_45 | Nervous | 0–2 | |
| CBCL_47 | Nightmares | 0–2 | |
| CBCL_50 | Fearful | 0–2 | |
| CBCL_53 | Overeating | 0–2 | |
| CBCL_54 | Overly tired | 0–2 | |
| CBCL_60 | Genital touching | 0–2 | |
| CBCL_66 | Repeats actions | 0–2 | |
| CBCL_68 | Screaming | 0–2 | |
| CBCL_71 | Self-conscious | 0–2 | |
| CBCL_73 | Sex problems | 0–2 | |
| CBCL_75 | Shyness | 0–2 | |
| CBCL_87 | Mood changes | 0–2 | |
| CBCL_88 | Sulks | 0–2 | |
| CBCL_96 | Sex thoughts | 0–2 | |
| CBCL_100 | Sleep problems | 0–2 | |
| CBCL_103 | Feels sad | 0–2 | |
| CBCL_109 | Whining | 0–2 | |
| CBCL_112 | Worries | 0–2 | |
| SocAnx_01 | Tolerate feelings of anxiety | -3–3 | **Extended Strengths and Weaknesses Assessment of Normal Behaviors** |
| SocAnx_02 | Handle fears of seeming anxious | -3–3 | Alexander et al., 2020 |
| SocAnx_03 | Stay relaxed in social situations | -3–3 | http://www.eswan.org |
| SocAnx_04A | Seek out social activities | -3–3 | |
| SocAnx_04B | Enjoy social activities | -3–3 | |
| SocAnx_05 | Control anxiety in social situations | -3–3 | |
| MFQ_01 | Feels miserable | 0–2 | **Mood and Feelings Questionnaire** |
| MFQ_02 | Enjoys nothing | 0–2 | Costello & Angold, 1988; |
| MFQ_03 | Less hungry | 0–2 | Burleson-Daviss et al., 2006; |
| MFQ_04 | Eating more | 0–2 | Sund et al., 2001; |
| MFQ_07 | Restless | 0–2 | Thabrew et al., 2018 |
| MFQ_10 | Indecisiveness | 0–2 | https://www.corc.uk.net/outcome-experience-measures/ |
| MFQ_12 | Talks less than usual | 0–2 | mood-and-feelings-questionnaire-mfq/ |
| MFQ_15 | Worries of future | 0–2 | |
| MFQ_21 | Concentration problems | 0–2 | |

**Table 2.** (Continued)

| Variable | Construct/ Content | Scale | Questionnaire and Literature |
|---|---|---|---|
| MFQ_22 | Worries of bad events | 0–2 | |
| MFQ_23 | Self-Hatred | 0–2 | |
| MFQ_27 | Feels lonely | 0–2 | |
| MFQ_28 | Feels unloved | 0–2 | |
| MFQ_29 | Has no fun at school | 0–2 | |
| MFQ_30 | Feels inferior to others | 0–2 | |
| MFQ_34 | Feels no joy when praised | 0–2 | |
| SCARED_GD | Generalized anxiety | 0–18 | **Screen for Anxiety Related Disorders** |
| SCARED_SC | Social anxiety | 0–26 | Behrens et al., 2019; |
| SCARED_SH | School avoidance | 0–8 | Birmaher et al., 1997, 1999 |
| SCARED_SP | Separation fears | 0–16 | |
| SDQ_03 | Head & stomach aches | 0–2 | **Strengths and Difficulties Questionnaire** |
| SDQ_06 | Preference for solitude | 0–2 | Achenbach et al., 2008; |
| SDQ_08 | Many worries | 0–2 | Goodman, 1997; |
| SDQ_16 | Nervous in new situations | 0–2 | Goodman et al., 2010; |
| SDQ_24 | Easily scared | 0–2 | Muris et al., 2003; Woerner et al., 2004 http://sdqinfo.org |
| SC_02C | Often annoyed | 0–1 | **HBN's Symptom Checker** |
| SC_03C | Loss of interests | 0–1 | http://childmind.org/symptomchecker/ |
| SC_08C | High sex interest | 0–1 | |
| SC_11C | Anxiety attacks | 0–1 | |
| SC_12C | Agoraphobia | 0–1 | |
| SC_14C | Fear of kidnapping | 0–1 | |
| SC_15C | Worries about harm to parents | 0–1 | |
| SC_17C | Trouble sleeping without parents | 0–1 | |
| SC_18C | Afraid to be in room without parents | 0–1 | |
| SC_19C | Nervous around people | 0–1 | |
| SC_20C | Nervous in front of others | 0–1 | |
| SC_21C | Unable to speak in front of others | 0–1 | |
| SC_22C | Specific phobias | 0–1 | |
| SC_23C | Often worries | 0–1 | |
| SC_25C | Unable to relax | 0–1 | |
| SC_26C | Intrusive thoughts | 0–1 | |
| SC_27C | Compulsions | 0–1 | |
| SC_30C | Weight worries | 0–1 | |
| SC_33C | Binge-eating | 0–1 | |

## Data preprocessing

All analyses were performed in R (version 4.3.1). First, missing data were imputed using a multivariate imputation algorithm ("mice" package in R). For ordinally scaled variables with few categories (i.e., two to six answer categories), classification and regression trees were used as the imputation method, for variables with many categories (i.e., upwards of 8 answer categories) random forests was utilized, and for variables with two categories, logistic regression [40]. We validated the imputation method through a simulation procedure. For the simulation, we randomly removed a percentage of data from the complete dataset, which corresponded to the mean percentage of missing data per observation in

the full dataset (~11.6%). These missing values were imputed with the methods described above, and the distributions between imputed and real data compared. The distributional overlap was satisfactory. The pertinent figures can be consulted in the "Imputation" section of the supplementary materials, along with further details on the procedure (Fig A in S1 Text). This approach ensured that the retained variables were psychometrically adequate and analytically robust. After imputation, the diagnostic category variables were dummy-coded. Followingly, a z-standardization was performed on all numerical variables except age. Then, the data were randomly stratified into two halves while controlling for age and sex, resulting in two comparable datasets for EFA and CFA (Dataset 1 (DS1) = 2,045 participants; Dataset 2 (DS2) = 2,097 participants).

## Exploratory factor analysis (EFA)

For the EFA, dataset 1 (DS1) was utilized. The suitability of the data for factor analysis was assessed through the calculation of the Kaiser-Meyer-Olkin (KMO) test [38]. Parallel analysis was applied to assess the number of factors to extract [41]. Oblique rotation was selected as the matrix rotation method, as is apt for correlated factors [42]. The model was calculated using the Mean and Variance Adjusted Weighted Least Squares (WLSMV) estimator. The selected fit indices and values to evaluate acceptable to good model performance were $RMSEA < 0.08 / < 0.06$, $SRMR < 0.08$, and CFI and $TLI > 0.90 / > 0.95$ [43,44]. All reported fit indices, unless otherwise noted, are scaled to keep them comparable between models. $\chi^2$ is reported but not considered as a main model evaluation criterion because of its sensitivity and therefore unreliableness with large sample sizes. Since fit indices alone may be insufficient to appraise model appropriateness in certain cases, the factor loading structure was analyzed for cohesion, as well as consistency with previous theories and findings on HiTOP's internalizing spectrum [45,46]. The internal consistency of the extracted latent factors was determined with Cronbach's α [47]. Finally, for model building, only the primary loadings, i.e., the highest variable-to-factor loadings, were considered for factor allocation. Approximately 35% of the variables had a second loading with $\lambda > 0.2$, which could have been added as a cross-loading. Nonetheless, to simplify the interpretability of the initial latent factors, and because patterns of co-occurrence could be implemented in later CFA steps, no cross-loadings were added.

## Confirmatory factor analysis (CFA)

The goal of the CFA was to verify the factor structure using dataset 2 (DS2) through an iterative process that tested various model variants, and incorporated modification indices. To ensure the aptitude of changes to the CFA model, ANOVAs were performed, and fit indices were evaluated. To further argue for the strong relationships between internalizing psychopathologies, we compared the resulting CFA model with correlated factors to a model with equal structure, but with an added constraint of uncorrelated factors.

## Bifactor and hierarchical modeling

The factor model was extended by exploring possible hierarchical and bifactor CFA structures. Additionally, the so-called "*bass-ackward*" method was implemented, as a growing body of literature argues for its relevance and accuracy in hierarchical analyses [48,49]. This method applies Principal Component Analysis (PCA) to find hierarchical components and removing redundant ones, i.e., components with high correlation (Pearson's r ≥ 0.9) and congruence (Tucker's φ ≥ 0.9) across all levels of the hierarchy [36,48]. Considering the results from parallel analysis and the EFA and CFA, we also adopted a four-component solution for the bass-ackward model. Following the recommendations by Forbes, the resulting model was fit as a hierarchical CFA and fit indices to the other models were compared [48]. The bass-ackward method was implemented utilizing the complete sample size to ensure sufficient power for analysis.

## Multigroup analysis

To assess the robustness of the factor structure, three separate multigroup analyses (MGA) were conducted, comparing model invariance for male (N = 2,635) and female (N = 1,507) participants, for prepubescent (N = 2,560) and postpubescent (N = 1,582) participants, and for participants with an internalizing diagnosis (N = 844), with an externalizing diagnosis (N = 2,744), and with no given clinical diagnosis (N = 554). Configural, metric and scalar invariance were considered as adequate indicators of measurement equivalence between subgroups [42,50]. Differences in |ΔRMSEA > 0.010| indicated measurement inequivalence [51].

For the age-based MGA, the mean age of puberty onset in the children and adolescents was used, resulting in a split of the sample at 10.84 years of age. This parent-reported puberty onset variable was taken from HBN's pre-interview questionnaire. For the diagnosis-based comparison, all participants with no diagnosable disorder, or with missing diagnoses due to missing data or incomplete clinical evaluations were grouped together into the "No Clinical Diagnosis" category, as they did not receive a formal clinical diagnosis. The internalizing group included the DSM categories of anxiety disorders, bipolar disorders, depressive disorders, feeding and eating disorders, gender dysphoric disorders, obsessive compulsive disorders, and traumatic stress disorders. In turn, the externalizing group was comprised of disruptive and impulse control disorders, elimination disorders, neurocognitive disorders, neurodevelopmental disorders, personality disorders, schizophrenic disorders and substance addiction disorders. The complete dataset (N = 4,142) was utilized for all MGAs to ensure large-enough subsample sizes. A fourth MGA combining the subsampling of sex and age groups, i.e., creating the subgroups of, e.g., postpubescent males or prepubescent females, was performed and is only reported in the supplementary materials (Table E in S1 Text).

## Structural equation modeling (SEM)

The purpose of SEM was expanding the model by assessing the relationship between the latent variables, as well as subsequently testing predictors of the latent variables. SEM was conducted with the full dataset of 4,142 participants. The structural model was built by testing different latent factor regressions, in part guided by existing literature on the causal processes between disorders. Further, the factor covariances from the measurement model (CFA) aided in the identification of potential regression paths to include in the model. This initiated an iterative process of model testing. In each structural model, the factor regressions were inspected. High regression scores, their statistical significance, the model's fit indices (CFI, TLI, RMSEA and SRMR), as well as ANOVAs between the models, if nested, attested the strength and validity of a relationship between latent factors. This way, relevant latent factor regressions to keep in the model were selected.

## Predictor and QoL indicator analysis

Beyond the latent factor relationships, nine sociopsychological predictors were added to the model. Their selection was guided by their availability in the HBN dataset and their relevance as identified in prior literature (e.g., parental behaviors and rearing practices, or peer victimization) [52–54]. The predictors encompassed constructs such as parents having a negative attitude towards their child, parental life dissatisfaction, peer bullying, mother involvement, inconsistent discipline, poor monitoring or corporal punishment by the parents, as well as a child or adolescent's age and sex. The variable content, scale and source questionnaires are listed in the supplementary materials (Table A in S1 Text). Piece-wise SEM (pwSEM) [55] was executed to analyze the effect of the predictors on the latent factors. Fit measures, effect sizes and the individual predictors' $R^2$ were compared between the models. In addition to exclusion of badly performing predictors through pwSEM, lasso regression was implemented to identify the best-performing predictors. Variable selection thus combined automated and criteria-based steps: poorly performing predictors were sequentially removed through the iterative SEM process, i.e., based on $p < 0.001$, $\beta > 0.1$, and lack of model fit improvement. Only variables that survived lasso regularization and met the significance and effect size thresholds were retained in the final model.

Finally, the WHO Disability Assessment Schedule's (WHODAS) sum score, which represents overall daily functioning in children and adolescents, was included as a QoL indicator from the four latent factors. This measure captures domains such as autonomy and self-care (e.g., being able to take care of their own hygiene), mobility (e.g., being able to walk a mile) and socio-emotional health and participation in society (e.g., controlling adverse feelings). After incorporating the sum score in the model, non-significant effects of the latent factors on daily functioning were removed again, retaining only those with a significance level of p < 0.001 and an effect size of $\beta > 0.1$.

## 3. Results

### Sample characteristics and aptitude of data

Sample characteristics are shown in Table 1. Age, sex, and race distributions remained similar between the complete sample and the two stratified halves (*DS1* and *DS2*). Approximately 80% of the sample had at least one diagnosis, 18.55% of which were internalizing psychopathologies. No statistically significant differences in age or sex were identified between the full and split datasets. Both halves showed comparable aptitude for factor analysis, with mean KMOs of 0.94.

### Exploratory factor analysis

Parallel analysis in DS1 produced a four-factor solution, explaining 47.6% of the total variance (CFI = 0.92, TLI = 0.91, RMSEA = 0.041; Table 3). Alternative factor extraction methods produced differing factor numbers. Only the four-factor solution is reported here, as it proved the most robust and statistically viable in subsequent analyses (i.e., did not have convergence issues). Details on alternative solutions and full loading matrices are provided in the supplementary materials (Table B, C in S1 Text).

The factors were labeled *Distress*, *Nervousness*, *Social Fears*, and *OC* (*Obsessions and Compulsions*). *Distress* comprised 27 variables (Cronbach's α = 0.91), with strongest loadings for *sadness*, *pessimism*, *self-hatred*, *apathy* and *worthlessness*. *Nervousness* included 23 variables (α = 0.91) such as *excessive worrying*, *fearfulness*, *generalized anxiety*, *anxiety attacks* and *separation fears*. *Social Fears* encompassed 13 variables (α = 0.89), including *enjoys social activities*, *relaxed in social situations*, *handles anxiety in social situations*, *preference for solitude* and *shyness*. *OC* summed 15 variables (α = 0.74) reflecting *obsessions*, *animosity*, *compulsions*, and *whining and crying*. This model is hereupon referred to as model "EFA". Fig 2 visualizes the loading structure; full details are provided in the supplementary (Fig B in S1 Text; Table C in S1 Text).

### Confirmatory factor analysis

The four-factor model was successfully validated in a separate dataset (DS2), with all loadings remaining significant (p-value < 0.001) and acceptable fit indices (model "CFA i"; cf. Table 3). Model fit improved after adding theoretically justified residual covariances (model "CFA ii"), while constraining factor correlations (model "CFA iii") significantly worsened fit. Following, model *CFA ii* was retained for subsequent analyses. ANOVAs between the CFA models confirmed the choice for this model. All latent factor covariances of model *CFA ii* had a p-value < 0.001. The highest standardized covariances were $\varphi$(*Distress, OC*) = 0.70, $\varphi$(*Distress, Nervousness*) = 0.67, and $\varphi$(*Nervousness, OC*) = 0.64, whereas the lowest one was $\varphi$(*Social Fears, OC*) = 0.37.

### Bifactor and hierarchical modeling

Alternative CFA model structures, such as CFA bifactor, hierarchical and second-order hierarchical models, were computed (Table 4). The hierarchical and second order models showed similar fit to model *CFA ii*. Although the bifactor model achieved marginally better fit (ΔRMSEA = 0.002 relative to *CFA ii*), the improvement was not considered substantial. Given the risk of overfitting, the more parsimonious correlated-factor model was preferred. This is further elucidated in the

**Table 3. Scaled Fit Measures of all Factor and Structural Models, and Multigroup Analyses.** The last column refers to the p-value of RMSEA and describes the certainty of the model's "close fit", i.e., the certainty of the RMSEA value. df = degrees of freedom; CI = confidence interval; Models: EFA = initial model with four factor structure; CFA i = initial CFA model replicating the EFA factor structure; CFA ii = CFA model with added modification index (residual variable covariance); CFA iii = model with CFA ii structure but uncorrelated factors; Sex MGA = sex-based MGA comparing male and female observations; Age MGA = age-based MGA comparing prepubescent and postpubescent observations; Diagnosis MGA = MGA based on diagnosis group, comparing observations with an internalizing or externalizing DSM diagnosis, or no formal clinical diagnosis. SEM i = model with selected predictors that survived pwSEM and lasso regularization; SEM ii = model with predictors with p < 0.001 and $\beta \geq 0.1$; SEM iii = model with predictors of SEM ii and added the indicator variable of daily functioning. *** = p-value < 0.001.

| Model | n | Fit Measures | | | | | |
| --- | --- | --- | --- | --- | --- | --- | --- |
| | | df | $\chi^2$ | CFI | TLI | RMSEA [CI] | p-value |
| EFA | 2,045 | 2,697 | 11,885.12*** | 0.92 | 0.91 | 0.041 [0.040;0.042] | >0.99 |
| CFA i | 2,097 | 2,919 | 18,007.99*** | 0.88 | 0.87 | 0.050 [0.049; 0.050] | 0.79 |
| CFA ii | 2,097 | 2,918 | 14,940.51*** | 0.90 | 0.90 | 0.044 [0.044;0.045] | >0.99 |
| CFA iii | 2,097 | 2,924 | 48,451.02*** | 0.62 | 0.61 | 0.086 [0.86;0.087] | <0.001 |
| Sex MGA | | | | | | | |
| Male | 2,653 | 2,918 | 18,267.79*** | 0.9 | 0.9 | 0.045 [0.044;0.045] | >0.99 |
| Female | 1,507 | 2,918 | 11,407.55*** | 0.90 | 0.90 | 0.044 [0.043;0.045] | >0.99 |
| Age MGA | | | | | | | |
| Prepub. | 2,560 | 2,918 | 16,320.45*** | 0.91 | 0.91 | 0.042 [0.042;0.043] | >0.99 |
| Postpub. | 1,582 | 2,918 | 11,645.63*** | 0.91 | 0.91 | 0.043 [0.043;0.044] | >0.99 |
| Diagnosis MGA | | | | | | | |
| Internalizing | 844 | 2,842 | 8,358.65*** | 0.87 | 0.87 | 0.048 [0.047;0.049] | 0.99 |
| Externalizing | 2,744 | 2,842 | 18,743.65*** | 0.90 | 0.89 | 0.045 [0.045;0.046] | >0.99 |
| No Clinical Diagnosis | 554 | 2,842 | 4,539.56*** | 0.93 | 0.93 | 0.033 [0.031;0.035] | >0.99 |
| SEM i | 4,142 | 2,919 | 30,399.07*** | 0.89 | 0.89 | 0.048 [0.047;0.048] | >0.99 |
| SEM ii | 4,142 | 3,226 | 30,612.45*** | 0.87 | 0.88 | 0.045 [0.045;0.046] | >0.99 |
| SEM iii | 4,142 | 3,306 | 30,884.56*** | 0.87 | 0.88 | 0.045 [0.044;0.045] | >0.99 |

discussion section. A complementarily (extended) bass-ackward analysis largely reproduced the same four-factor structure of model *CFA ii* with minor variable reassigments (< 5%), but the character of the factors remained unchanged (Table D in S1 Text). This model identified an overarching factor at the top linking *Distress* and *Fear*. *Fear* separated into *Nervousness* and *Social Fears*, whereas *Distress* remained intact, but was related to a fourth factor, *OC*, at the lowest level of the hierarchy (see Fig 3). However, higher-order models based on this structure failed to meet recommended fit thresholds (RMSEA < 0.05; see Table 4) and were not pursued further.

## Multigroup analysis

Measurement invariance across sex, age, and diagnostic groups was tested using multigroup CFA based on model *CFA ii*. Fit indices were similar across groups (female > male, postpubescent > prepubescent), with ΔCFI and ΔRMSEA values below 0.01, indicating configural, metric and scalar invariance (Table 3). Model fit was descriptively best in participants without clinical diagnoses, which must be interpreted with caution due to sample size variation and method effects. Implications of all MGAs can be found in the Discussion section. Configural, metric and scalar fit of all three MGAs are displayed in Table E in S1 Text.

## Structural equation modeling

A structural model including all latent factor regressions was estimated (e.g., *Distress* being predicted (~) by *Nervousness*, *Social Fears* and *OC*). Due to model complexity and bidirectional effects, the model could not converge. Thus, effects

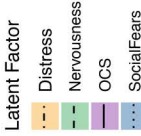

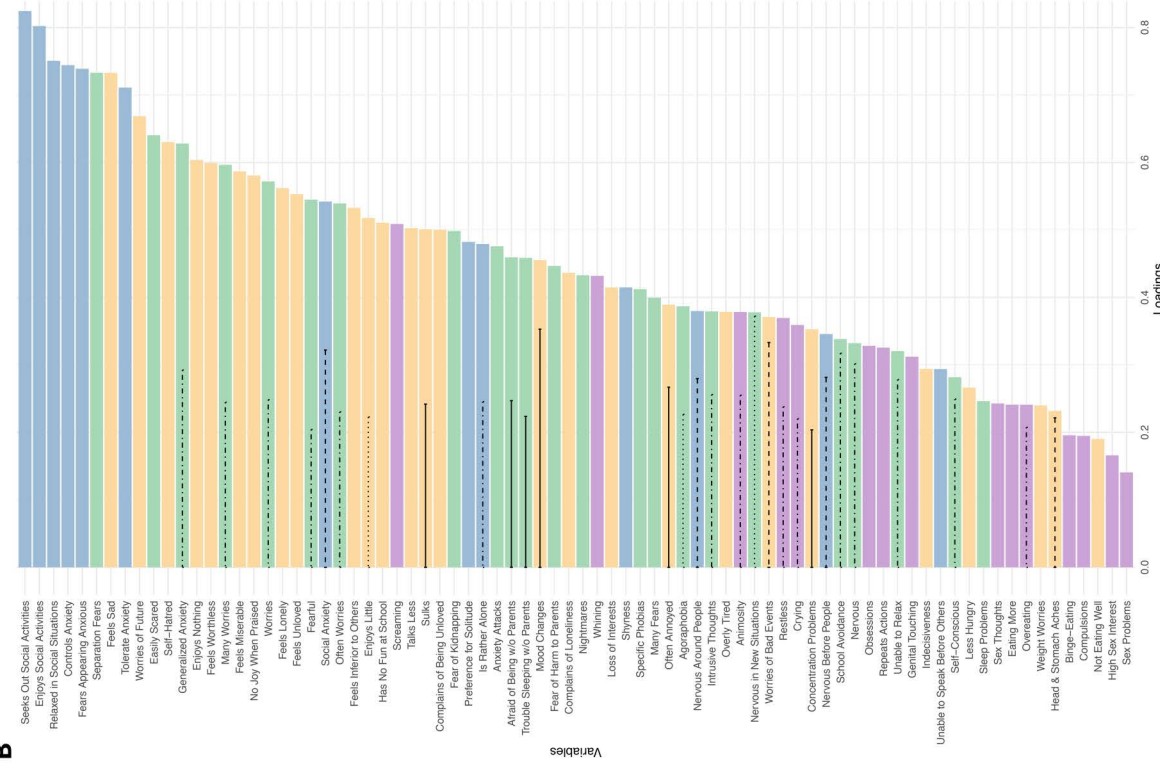

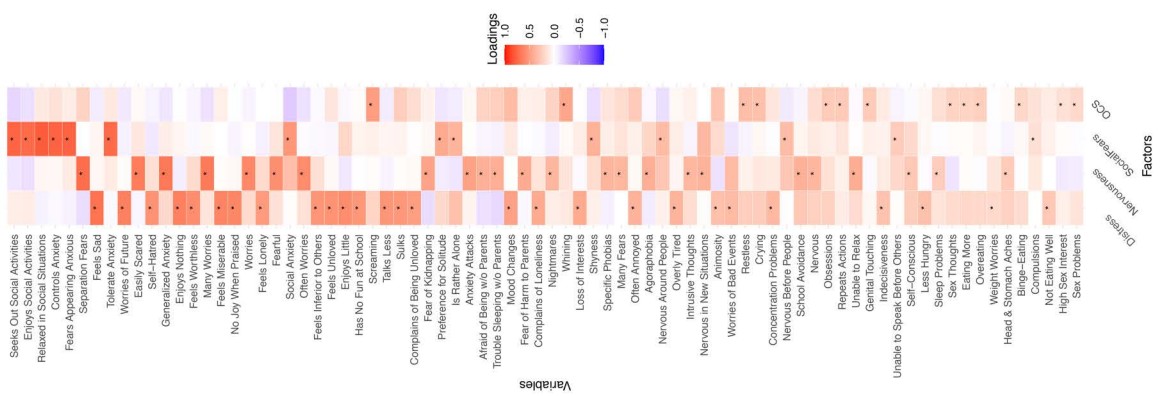

**Fig 2. Factor Loading Structure of Model *EFA*.** A. Heatmap of the latent factor loadings across all variables for *Distress*, *Nervousness*, *Social Fears*, and *OC* in descending order of loading size. The highest loading per variable is marked with an asterisk (*). B. Bar plot of the first and second highest loadings per variable, ordered by descending loading size. The primary, i.e., highest loading, is indicated by the color of the bars, the second loading

by the dashed, dotted or straight lines, e.g., the primary loading for "Enjoys little" is on *Distress*, whereas the second-highest loading is on *Social Fears*. Only second loadings with $\lambda \geq 0.2$ are present. The lower the primary loading, the more second loadings with $\lambda \geq 0.2$. The variable contents can be seen on the y-axis, and additional information consulted in Table A in <u>S1 Text</u>. *OC* = Obsessions and Compulsions; w/o = without.

**Table 4. Scaled Fit Measures of all Hierarchical and Bifactor models.** CFA ii is the standard correlated-factors model. The bifactor model assumes that all variables load directly onto a general factor and unto a specific factor, i.e., the model's four latent factors. In comparison, the hierarchical model has a higher-order factor above the four latent factors that explains the correlations among the latent factors. Variables do not directly load unto the higher-order factor. Analogously, the second order model is a form of hierarchical model, with OC being subjugated under *Distress*, given their high correlation (and regression effect in SEM; see below). The Bass-Ackward models depict the hierarchical structure proposed by the *extended* Bass-Ackward method. The higher-order model contains the relationships of an overarching internalizing factor to subjugated *Distress* and *Fear* factors, whereas the second-order model contains the division of *Fear* into *Nervousness* and *Social Fears*, and the inclusion of a further factor, *OC*, under *Distress*. *Fear* is not a first-order factor itself, i.e., does not directly load unto manifest variables. The p-value in the last column refers to the RMSEA value (test of close fit). All values are scaled.

| Model | n | Fit Measures | | | | | |
|---|---|---|---|---|---|---|---|
| | | df | $\chi^2$ | CFI | TLI | RMSEA [CI] | p-val. |
| **CFA** | | | | | | | |
| CFA ii | 2,097 | 2,918 | 14,940.51 | 0.90 | 0.90 | 0.044 [0.044;0.045] | >0.99 |
| Bifactor | 2,045 | 2,846 | 13,142.32 | 0.91 | 0.91 | 0.042 [0.041; 0.042] | >0.99 |
| Hierarchical | 2,045 | 2,920 | 15,328.49 | 0.90 | 0.89 | 0.045 [0.044; 0.046] | >0.99 |
| Second-order | 2,045 | 2,920 | 15,117.53 | 0.90 | 0.90 | 0.044 [0.044; 0.045] | >0.99 |
| **Bass-Ackward** | | | | | | | |
| Higher-order | 4,142 | 2,920 | 39,449.12 | 0.85 | 0.85 | 0.055 [0.054; 0.055] | 0 |
| Second-order | 4,142 | 2,921 | 39,431.20 | 0.85 | 0.85 | 0.055 [0.054; 0.055] | 0 |

were first examined iteratively and then integrated into a final model (*SEM i*), which achieved good fit (<u>Tables 3</u>, <u>5</u>). All regression scores for the tested structural models are listed in supplementary Table F in <u>S1 Text</u>.

## Predictor and QoL indicator analysis

Subsequent predictor analyses via piecewise SEM and lasso regression identified key predictors (e.g., sex, parental negative attitude, peer bullying, parental life dissatisfaction, inconsistent discipline; Model "SEM i"). Not all variables remained significant or with $\beta \geq 0.1$ (cf. Table G, H in <u>S1 Text</u>), and as such were removed from the model once again. A retested model (Model "SEM ii") reached acceptable fit indices, and was supplemented with a QoL indicator from the WHODAS, representing daily functioning. This last model (Model "SEM iii"; <u>Fig 4</u>) established significant influences of *Distress*, *Social Fears* and *OC*, but not *Nervousness*, unto daily functioning, and abided by Hu and Bentler's proposed fit index thresholds for acceptable to good model fit. The variance explained by the predictors for *Distress*, *Nervousness*, *Social Fears* and *OC* were 21%, 13%, 7%, and 20%, respectively.

## Clinical plausibility

To assess clinical validity, latent factors scores were compared across diagnostic groups, i.e., the type of diagnosis and the diagnostic categories of the participant's diagnoses were compared with their predicted latent factor scores (<u>Fig 5</u>). As expected, the internalizing group exhibited significantly higher scores for all four factors compared to the two other groups

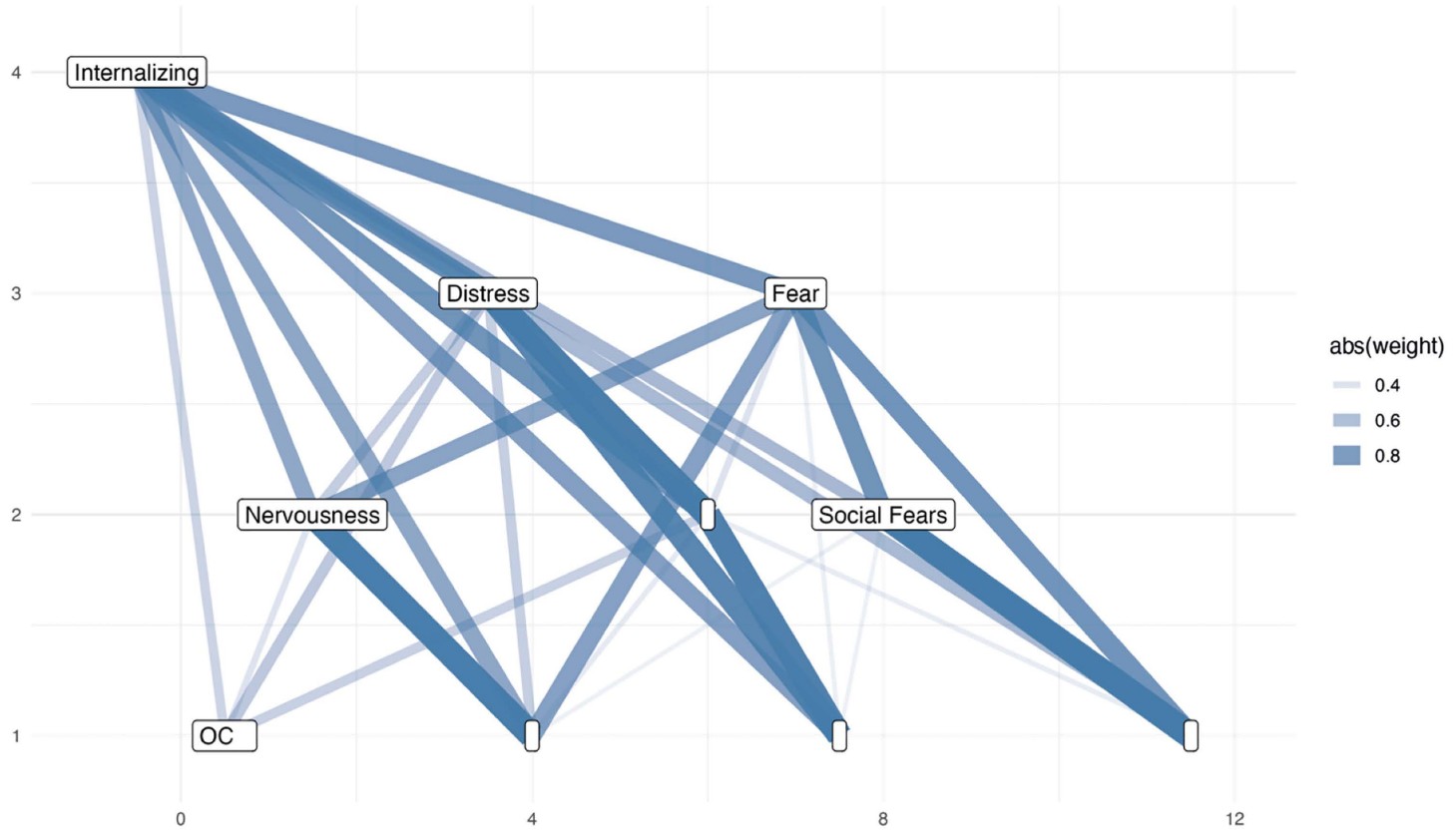

**Fig 3. Hierarchical Component Structure.** This structure was modeled implementing the Extended Bass-Ackward Method on the complete dataset. This model includes the thresholds of Pearson correlation r ≥ 0.9 and Tucker's congruence coefficient φ ≥ 0.9 to ensure the removal of redundant components. The correlations indicate the relatedness of the individual participant's component scores, whereas the congruence assesses the similarity of the variable loadings for all components. Based on the set correlation and congruence thresholds for the components across levels, the extended Bass-Ackward method recommended the removal of four components, indicated by empty labels in the figure. abs(weight) = absolute regression weights, i.e., loadings across levels.

**Table 5. Regression Coefficients for Final SEM.** The final model (Model SEM i) selected for further analyses. The p-value refers to the significance of the regression scores. All further structural equation models can be consulted in Table F in S1 Text. OC = Obsessions and Compulsions.

| Regression effect | Regression score | p-value | Fit Measures |
|---|---|---|---|
| **Nervousness ~ Distress** | 0.68 | < 0.001 | **CFI**: 0.89 |
| **Social Fears ~ Nervousness** | 0.51 | < 0.001 | **TLI**: 0.89 |
| **OC ~ Distress** | 0.53 | < 0.001 | **RMSEA**: 0.0 [0.047; 0.048] |
| **OC ~ Nervousness** | 0.26 | < 0.001 | **SRMR**: 0.08 |

([Fig 5A](link)). Within diagnostic categories, *Distress* was most elevated in the depressive disorders, *Nervousness* in anxiety disorders, and *OC* in obsessive-compulsive-related and depressive disorders. Depressive and anxiety disorders had the highest predicted *Social Fears* latent factor score. In general, categories of internalizing disorders mapped higher unto the latent factor scores than those of externalizing disorders ([Fig 5B](link)).

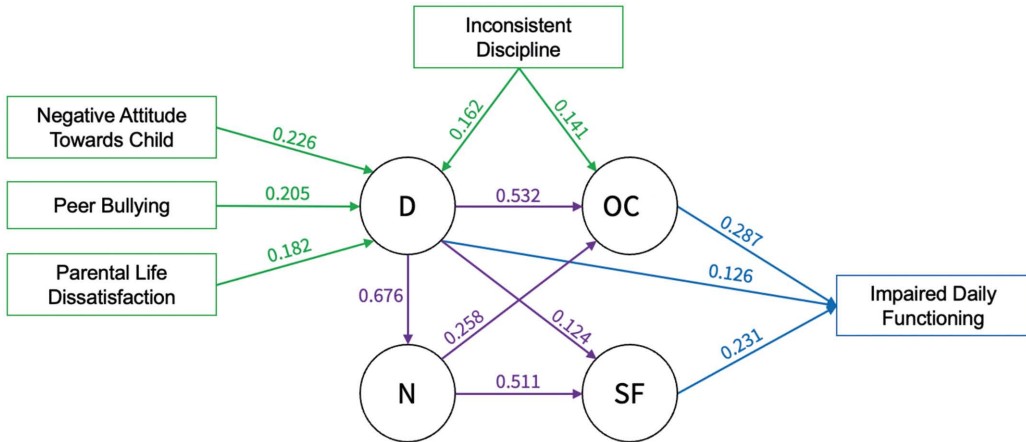

**Fig 4. Final Model.** Model *SEM iii* with latent factors, regression effects between latent factors in purple, predictors in green and the QoL indicator in blue. All effect sizes are standardized and have a p-value < 0.001. D = *Distress*; N = *Nervousness*; OC = *Obsessions and Compulsions*; SF = *Social Fears*.

Adapted CBCL sum scores (excluding overlapping CBCL variable utilized in the factor analysis were correlated with the latent factor scores (Fig 6). *Nervousness* showed the strongest association with the CBCL Anxiety-Depression scale, whereas *Social Fears* and *Distress* demonstrated the strongest relationship with the Withdrawal-Depression scale. No CBCL sum score directly reflects obsessive-compulsive or sexual symptomatology, thus, the *OC* factor was not strongly associated with any dimensional sum score. Overall, externalizing dimensions (e.g., Attention Problems and Rule-Breaking Behavior) corelated less strongly with the latent factors, except for Aggressive Behavior. Implications of these results are reviewed in the Discussion section.

## 4. Discussion

This study aimed to replicate HiTOP's structure of transdiagnostic internalizing psychopathology for childhood and adolescence. An alternative, four factor structure was discovered, which was invariant to sex and age. The resulting dimensional model was compared with traditional DSM diagnoses and the CBCL diagnostic tool, further supporting the plausibility and coherence of the identified latent factors.

To gain a more comprehensive understanding of internalizing pathologies predictors for and one QoL indicator of the factors were examined. Key predictors included experiences of inconsistent discipline, peer bullying, and parental life dissatisfaction, as well as parents having a dysfunctional or negative attitude towards their children. Importantly, individuals with higher scores on the *OC*, *Social Fears* and *Distress* factors exhibited greater impairments in daily functioning.

Based on the variables associated with the four latent factors, the labels of "*Distress*", "*Nervousness*", "*Social Fears*" and "*OC*" (*Obsessions and Compulsions*) were assigned. Please note that in the following section, "latent factors" (or "factors") are to be understood as the four factors resulting from the factor analysis in this work, whereas "subfactors" refer to those ascertained by the HiTOP consortium in their hierarchical model. The *Distress* and *Nervousness* latent factors closely resembled HiTOP's *Distress* and *Fear* subfactors, respectively [3]. For example, both the *Distress* latent factor and subfactor include constructs such as dysphoria, anhedonia, suicidality, and irritability, and both *Nervousness* and *Fear* are associated with specific fears, phobias, and anxiety. The overlap between our *Distress* latent factor and HiTOP's *Distress* subfactor was considerable, and we thus opted to adopt the HiTOP label. Contrarily, since HiTOP's *Fear* includes a broader range of anxiety-related components than our *Nervousness* factor, these labels were not aligned. Further, unlike the HiTOP model, where *Fear* also includes traits of social and performance anxiety, our factor model found these to be

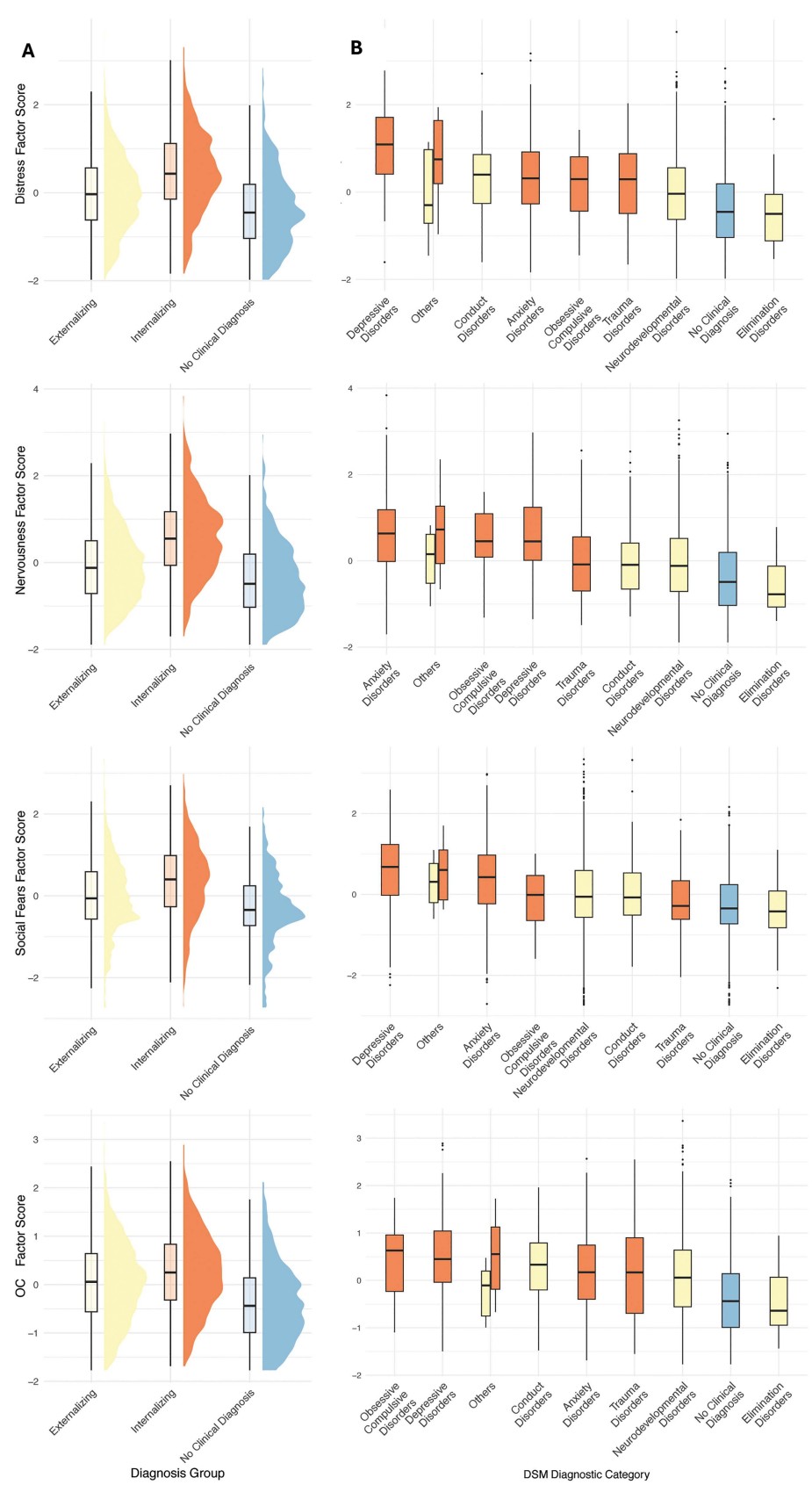

**Fig 5. Clinical Plausibility of Latent Factors.** A Mapping of the latent factors with the type of diagnosis. Internalizing disorders had the highest score on all latent factors, followed by externalizing disorders, and no clinical diagnosis. These differences were all statistically significant. B Mapping of the latent factors with diagnostic categories of DSM. The category of "Others" included diagnostic categories with fewer than 10 observations. Categories included are *Bipolar and Related Disorders* (n = 7), *Feeding and Eating Disorders* (n = 3), *Gender Dysphoria* (n = 2), *Neurocognitive Disorders* (n = 4), *Schizophrenia Spectrum and other Psychotic Disorders* (n = 5) and *Substance and Addictive Disorders* (n = 3). The low sample sizes may lead to the latent factor score of these not to be representative. Yellow = Externalizing disorders; Orange = Internalizing disorders; Blue = No clinical diagnosis.

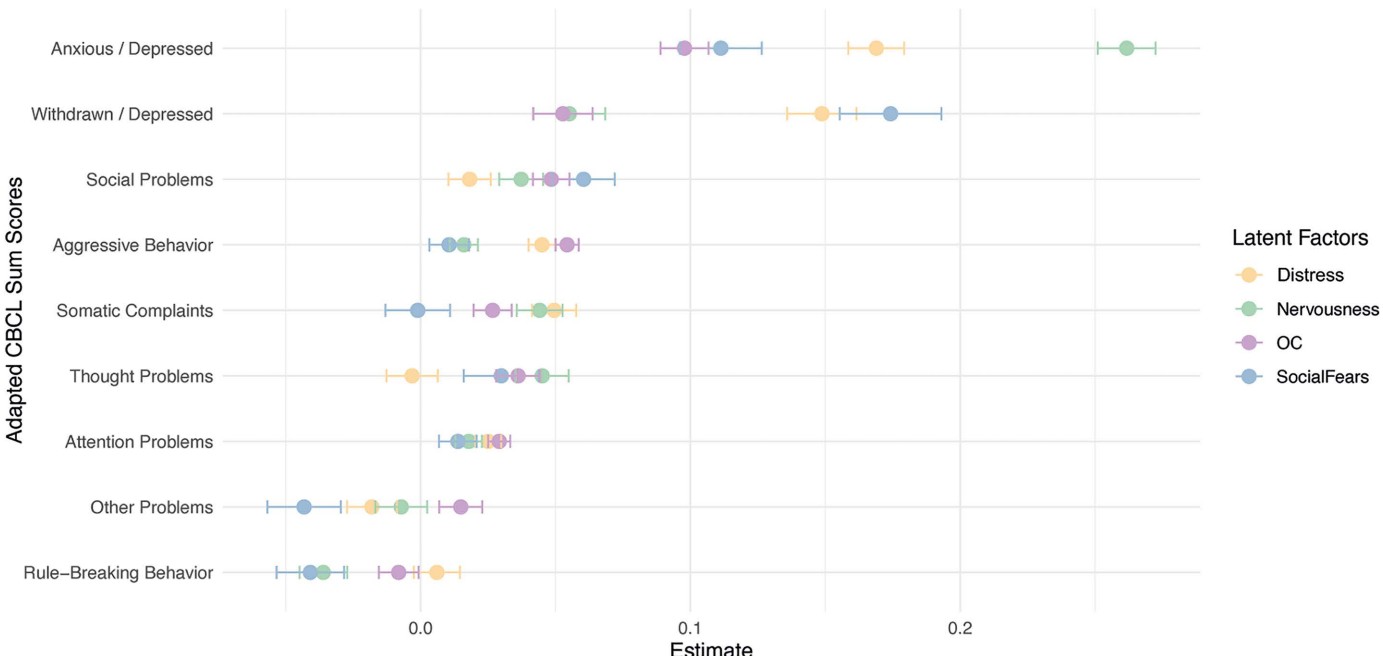

**Fig 6. Diagnostic Plausibility of Latent Factors.** Mapping of latent factor scores with adapted CBCL sum scores. Here, the CBCL sum score of "Other Problems" refers to symptom and trait domains such as nail biting, being cruel to animals, increased sleep, and enuresis, among others.

distinct from all other anxious and nervousness symptoms, forming a cohesive latent factor labeled *Social Fears*. Components of obsessions, checking, and rituals are distributed between *Distress* and *Fear* in the HiTOP, whereas in our model, they were combined within the *OC* factor. This factor included further behavioral disinhibitions such as the touching of one's own genitals, binge-eating, and excessive screaming, whining, and crying. Further nuances emerged with food-related variables: variables related to eating less than usual or weight worries were allocated to *Distress*, whereas variables on food disinhibitions and eating more or uncontrollably were allocated within the *OC* factor. This indicates potentially separate domains of eating pathologies that may evolve differently across development.

The content of the four factors also closely aligns with the diagnostic criteria for Major Depressive Disorder, General Anxiety Disorder (GAD), Social Anxiety Disorder (SAD), and Obsessive-Compulsive Disorder (OCD) in the DSM-5. For example, GAD in children and adolescents includes worries about the safety of parents and guardians, and physical manifestations of anxiety such as head or stomach aches; features that are also encompassed within the *Nervousness* factor. Nonetheless, the latent factors extend beyond the dichotomous, categorical boundaries of the DSM-5 diagnoses. In fact, our analyses revealed covariances between the latent factors, which mirror well-documented patterns of comorbidity in the literature. For example, the strong association between mood and anxiety disorders, as reflected in our *Distress* and *Nervousness* factors, is consistent with findings of high pediatric comorbidity [56]. The progression from depressive

to anxiety disorders, and vice versa, has been observed [57]. Similarly, anxiety disorders are often highly comorbid with each other: In a sample of Norwegian youth with SAD, 58.5% were found to meet the criteria for GAD, and 25.7% that for specific phobias [58]. These results provide an explanatory approach for the strong association between *Nervousness* and *Social Fears*. In addition, both depressive and anxiety disorders have been linked to OCD [59–61]. According to the DSM-5 diagnostic criteria obsessive-compulsive symptoms often "cause marked anxiety or distress", thus explaining the distress-related symptoms (e.g., crying, whining and screaming) identified within the *OC* factor. This connection underscores the overlap between the *OC* factor and the broader *Distress* and *Nervousness* factors, further highlighting the interconnected nature of these psychopathologies[8]. Tentatively, the addition of cross-loadings would have further modeled these patterns of comorbidity, albeit we opted to exclude them from our model to facilitate the interpretation of the latent factors, and because comorbidity was further integrated into the model through factor covariances and regressions.

In sum, our factors showed strong symptomatic coherence, high internal consistency, and many parallels to previous findings on co-occurrence and comorbidity. One further strength of such a model is the ability to characterize each individual on a dimensional level for each of the latent factors. This specifies a child or adolescent's symptom and trait manifestation within the internalizing domain, highlighting areas of gravity, as well as potential treatment targets. The model achieved satisfactory fit, and moreover, was found to be very robust, as its structure was confirmed through the analysis of a separate dataset with CFA. Despite the model's robustness, we deemed it important to inspect its validity for different subgroups within the sample – a valuable analytical step mostly overlooked in past literature. Various multigroup analyses showed invariance to sex and age, providing evidence that there were no sample effects, i.e., our sample including more male and prepubescent participants did not skew the factor model. The model also seems invariant to type of diagnosis, with it even being a good fit for participants whose primary diagnosis is externalizing. One possible interpretation is that the DSM diagnostic categories do not adequately capture the symptomatology and thus purvey a diagnosis whose diagnostic criteria are not fully or correctly met. Alternatively, it could be that too many children and adolescents receive externalizing diagnoses because those symptoms are more salient to parents and peers, although they also present clinically relevant internalizing symptoms and traits. This highlights the issue of the high rate of comorbidity across internalizing and externalizing disorders once again. Nonetheless, the model provided evidence of invariance and thus seems broadly generalizable to the subgroups that were inspected.

Generalizability is only one aspect of clinical plausibility. To further support the model's plausibility, we mapped the DSM's diagnostic categories onto the latent factor scores, revealing a strong overall alignment between the categories and the factors. For example, depressive disorders, anxiety disorders and obsessive-compulsive disorders mapped the highest on *Distress, Nervousness* and *OC*, respectively. The diagnostic categories of externalizing psychopathologies generally mapped the lowest on all four latent factors. Additionally, individuals with an internalizing pathology had consistently stronger scores on all four factors than those with an externalizing diagnosis and those with no clinical diagnosis at all. Taken together, these results highlight a high consistency between the proposed latent factors and existing clinical tools, further substantiating the model's clinical plausibility.

Our findings have practical implications for screening and intervention. The identified dimensional structure can inform the development of new screening tools that capture broad internalizing dimensions. Such tools could serve both clinical and non-clinical settings: For example, in schools they could facilitate early identification of individuals at elevated risk across domains. In clinical settings, dimensional scores could complement categorical classifications by providing a profile of severity and comorbity. Further, in intervention planning, a dimensional model could help tailor treatments to shared underlying processes, as well as to the individual. In this way, our study provides an empirical foundation for the eventual development of age-appropriate screening and diagnostic tools with clear translational value.

Finally, the exploration of possible correlates of the four latent factors yielded several notable findings. Among the predictors, only a few of the selected variables survived all levels of analysis and most of their relevance concentrated on *Distress*. This observation raises the possibility that *Distress* mediates the influence of these predictors on other latent

factors, given its substantial covariances with and regression effects on *Nervousness* and *OC*. The retained predictors in the model align with previous research. For example, peer bullying (or "victimization") has consistently been linked to depression and to internalizing psychopathology altogether [62]. Similarly, parental discipline (or "dysfunctional parenting") has shown strong associations with both depression and OCD [63–65]. Furthermore, impairments in various aspects of daily functioning have been reported in children with depressive, social anxiety and obsessive-compulsive symptoms, agreeing with our findings of the impact of *Distress*, *Social Fears* and *OC* on daily functioning [66–69].

## Limitations and future research

As with any factor analysis, the results of this work are influenced by the included variables, i.e., their content and informants. Regarding informants, the present study included variables based on parental reports, which might introduce a systematic bias [70,71]. Externalizing symptoms are often more salient and therefore more readily reported by parents, whereas internalizing symptoms typically require greater sensitivity and may be underestimated. The under-reporting of internalizing symptoms should be considered when interpreting our findings. Nonetheless, the imbalance in parents-reports between externalizing and internalizing domains is less of a problem here, since we only included internalizing symptoms. Considering variable content, studies examining the internalizing spectrum in adults have identified varying factorial structures depending on the included constructs (for an overview, see [30]). Our selection of pertinent variables to include in the factor analysis followed a data-driven and a content-driven approach. The former is purely objective, since it removes variables statistically unsuitable for analysis, e.g., the child-reported variables had mostly >80% missing data and were thus deemed non-imputable. The latter subsumes more subjective decisions, as the exclusion or inclusion of a variable was ultimately (hu)man-made but we tried to make this process as literature driven as possible: We followed the indications of the literature on each variable's questionnaire concerning the constructs that the variable is intended to measure, and matched these with the constructs included in HiTOP's internalizing spectrum. For example, if a questionnaire's documentation explicitly stated the intent of measuring externalizing symptoms, we discarded it.

Adding to the limitation specified at the very beginning of this section, due to variable selection and availability, our model does not entail all domains relevant to internalizing psychopathology. Specifically, trauma and stressor-related disorders were excluded due to insufficient variable representation or excessive missing data within our sample. On top of that, alternative model conceptualizations to ours may provide an equally valid or even better representation of pediatric internalizing psychopathology. To address this, we explored three alternative factor model structures commonly proposed in the literature: the bifactor and hierarchical models, as well as a modified hierarchical model with *Distress* positioned hierarchically over *OC*. Beyond factor analysis, we additionally implemented the extended bass-ackward hierarchical approach. While the alternative CFA models demonstrated adequate fit, it is difficult to determine whether they are as equally suitable as our selected model. For example, the additional variance captured by a bifactor may not be theoretically meaningful (i.e., is residual covariance) and "only a statistical summary of the covariance among psychopathological symptoms" [72]. In other words, the general factor in bifactor models has a propensity to reflect residual covariance among items, that albeit statistically strong is theoretically ambiguous. As a result, it may represent a statistical artifact rather than a meaningful dimension. Although model fit was an important consideration for our model selection, our evaluation also took theoretical interpretability and empirical support into account, since solely relying on model fit has been deemed inadequate for highly complex models in the past [73,74]. Ultimately, our model constitutes a robust solution congruent with literature and is in line with the principle of parsimony. Additionally, the results from the bass-ackward analysis largely confirmed our variable-to-factor structure, presenting an internalizing psychopathological structure of *Distress*, *Nervousness*, *Social Fears* and *OC*, as well. The modeled hierarchical structure was cohesive and congruent with previous literature, e.g., the internalizing dimension being largely characterized by *Distress* and *Fear* is consistent with findings on the internalizing spectrum [75,76]. This approach also reflected our expectation that *OC* might partially load onto the *Distress* factor. Despite theoretical coherence, the hierarchical models derived from the bass-ackward analysis did not

yield satisfactory fit indices, limiting the strength and interpretability of conclusions regarding the higher-order structure. In addition, computational limitations in subsequent MGAs and SEM prevented us from implementing the full three-tier hierarchical model. Nonetheless, we considered it essential to include an alternative, well-established hierarchical modeling approach in our analyses. Replicating findings across different analytical frameworks fosters comparability with previous work and highlights the importance of testing dimensional structures across complementary analytic strategies. Future research should continue to investigate the hierarchical organization of internalizing symptoms in children and adolescents to clarify the developmental validity of dimensional models like HiTOP.

The results of the multigroup analyses provided a basis to assume model invariance to the subgroups we tested for. Nonetheless, there were considerable differences in sample size between the different subgroups, which may lead to somewhat unstable fit indices as well as model non-convergence. To remediate this issue, we performed sample-size matched MGAs, which delivered similar results to the standard ones. We thus cautiously concluded that our results seem plausible, and our model is invariant, but further research must confirm them. For the diagnostic group comparison, the superior fit of the group with no clinical diagnosis may have arisen from the inclusion of the categories of participant's that had no diagnosis due to an incomplete evaluation or due to missing data, which, in reality, could possibly receive a diagnosis if the assessments were to be completed. Nonetheless, it could also be a methods effect due to the group's small sample size.

A further limitation is the labeling of our factors. We adopted HiTOP's *Distress* label for one of our latent factors due to high overlap of its associated psychopathological constructs. Our other factors did not directly align with HiTOP's, as the model does, for example, not include a factor explicitly covering obsessions, compulsions, behavioral disinhibitions and sexual fixations such as our *OC* factor. To this end, we adopted our own labels for three of the latent factors, suggesting that the structure of psychopathology in childhood may differ to that in adulthood.

Finally, one must consider the chosen predictors. We found parental life dissatisfaction and peer bullying, among other predictors, to be greatly associated with internalizing psychopathologies. Yet, many more sociopsychological variables not considered in this study due to lack of data availability surely influence internalizing psychopathology and should be examined in future longitudinal investigations. Nonetheless, the effects found show great potential for therapeutic targets and interventions, as well as early identification of children and adolescents at risk of developing internalizing psychopathologies.

Further recommendations for the future are the analysis of HiTOP model for other age groups, such as for elders, as well as for the whole dimensional psychopathological spectrum, instead of only focusing on the internalizing domain. This way, one could effectively test the hierarchical and transdiagnostic dimensional structure at different age ranges. In addition, variables from dimensional behavioral questionnaires may be more apt for this kind of investigation. The inclusion of non-behavioral variables, such as neuroanatomical or functional data, could reinforce the veracity of the model. For this, big data approaches should continue to be implemented to ensure well-powered analyses.

## 5. Conclusion

The findings of this study argue for an alternative understanding of dimensional internalizing psychopathologies in children and adolescents by establishing a robust four-factor structure. These factors not only align with DSM diagnostic categories, but also offer deeper insights into symptom co-occurrence patterns, exemplified by the distinct composition of the *OC* factor. Importantly, the model showed invariance across sex and age, suggesting its generalizability. Behavioral correlates, such as inconsistent discipline and peer bullying, were identified as predictors, particularly within the *Distress* factor, reinforcing findings from prior research and identifying actionable targets for intervention. The findings underscore the clinical plausibility of the proposed model. Taken together, the findings extend the understanding of pediatric psychopathology while addressing the complexity of the internalizing symptom and trait domain and add nuance to the dimensional conceptualization of psychopathology at the symptom and factor level. This work contributes to the growing body of

literature advocating for transdiagnostic and dimensional approaches to psychopathology, specifically focusing on childhood and adolescence, which will hopefully lead to improved interventions and mental health outcomes in youth.

## Supporting information

**S1 Text. Table A**. Study variables. Overview of variables, scale, and source questionnaires for all variables utilized in FA, MGA, and SEM. **Table B**. Fit Indices of Three, Four and Eight Factor Solutions. Fit indices of models with an alternative number of factors, i.e., the 3- and 8-factor solutions, are presented here. **Table C**. Factor Loading Structure of the Four Factor Model. Raw factor loadings of all variables to the factors in the four factor solution of the EFA, in addition to all variable communalities. **Table D**. Variables with Different Factor Allocation in CFA and Bass-Ackward Analysis. Differences between the CFA and Bass-Ackward method in regards to variable-to-factor allocation. **Table E**. Additional Fit Indices for MGAs. Reporting of additional fit indices of the MGAs reported in the main manuscript, i.e., strict invariance indices, and fit indices of an additional MGA looking at sex and age combined subgroups (e.g., "male-prepubescent" or "female-postpubescent"). **Table F**. Regression Effects in SEM. Standardized regression scores between latent factors for all structural models in SEM. **Table G**. Standardized Regression Coefficients in Piece-Wise SEM of Predictor (Indicator) Analysis. Standardized regression coefficients of the predictor selection process utilizing piece-wise SEM. **Table H**. Lasso Regression of Predictor (Indicator) Analysis. Standardized regression coefficients of the predictor selection process utilizing lasso regression. **Fig A**. Imputation of Simulated and Real Missing Values. Density and distribution curves of imputed simulated missing values, imputed real missing values and datasets with missing values and imputed values. **Fig B**. Path Diagram of the Four Latent Factor Solution. Path diagram with variable-to-factor overview corresponding to the factor loading structure of the four factor solution.
(DOCX)

**S1 Checklist. Questionnaire references (Table 2).**
(DOCX)

## Author contributions

**Conceptualization:** Ines M. Engler, Nicolas Langer.

**Formal analysis:** Ines M. Engler.

**Investigation:** Ines M. Engler, Nicolas Langer.

**Methodology:** Ines M. Engler, Nicolas Langer.

**Supervision:** Nicolas Langer.

**Visualization:** Ines M. Engler.

**Writing – original draft:** Ines M. Engler.

**Writing – review & editing:** Nicolas Langer.

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
