## [Decision Letter · Decision Letter 0]

21 Aug 2025

PMEN-D-25-00214

Toward Dimensional Psychiatry in Youth: A Data-Driven Analysis of Transdiagnostic Internalizing Symptoms in Childhood and Adolescence

PLOS Mental Health

Dear Dr. Engler,

Thank you for submitting your manuscript to PLOS Mental Health. After careful consideration, we feel that it has merit but does not fully meet PLOS Mental Health’s publication criteria as it currently stands. Therefore, we invite you to submit a revised version of the manuscript that addresses the points raised during the review process.

We look forward to receiving your revised manuscript.

Kind regards,

Juan Felipe Cardona, Ph.D.

Academic Editor

PLOS Mental Health

Journal Requirements:

i. State the initials, alongside each funding source, of each author to receive each grant.

ii. State what role the funders took in the study. If the funders had no role in your study, please state: “The funders had no role in study design, data collection and analysis, decision to publish, or preparation of the manuscript.”

2. Please ensure that your Ethics Statement is available in its entirety at the beginning of your Methods section, under a subheading 'Ethics Statement'.

3. We ask that a manuscript source file is provided at Revision. Please upload your manuscript file as a .doc, .docx, .rtf or .tex.

4. Please provide separate figure files in .tif or .eps format.

5. We have noticed that you have uploaded Supporting Information files, but you have not included a list of legends. Please add a full list of legends for your Supporting Information files after the references list.

Additional Editor Comments (if provided):

Reviewers' comments:

Reviewer's Responses to Questions

**Comments to the Author**

1. Does this manuscript meet PLOS Mental Health’s publication criteria?

Reviewer #1: Partly

Reviewer #2: Yes

2. Has the statistical analysis been performed appropriately and rigorously?

Reviewer #1: Yes

Reviewer #2: Yes

3. Have the authors made all data underlying the findings in their manuscript fully available (please refer to the Data Availability Statement at the start of the manuscript PDF file)?

Reviewer #1: Yes

Reviewer #2: Yes

4. Is the manuscript presented in an intelligible fashion and written in standard English?

Reviewer #1: Yes

Reviewer #2: Yes

Reviewer #1: Your use of a large clinical sample, robust statistical techniques (EFA, CFA, SEM, multi-group analysis), and alignment with HiTOP principles is commendable. However, the manuscript requires major revisions before it can be considered for publication. Below are specific comments to guide your revisions:

1. Strength: You clearly justified the use of factor analysis and model testing across independent datasets. Your statistical approach is rigorous and appropriate, including multi-group invariance testing, the extended Bass-Ackward method, and SEM with regularization.

Weakness: Please clarification of variable selection and handling of missing data is needed in the main text. Please briefly explain why 78 variables were retained and summarize the imputation strategy and missingness pattern.

2. Conceptual clarity: Given the age of the participants, the inclusion of “sexual tendencies” without context may be misinterpreted. We strongly encourage revisiting this label and providing clarification on how these symptoms were defined, measured, and interpreted (OCS-Obsessions, Compulsions, and Sexual Tendencies).

3. Practical Implication: How could your model be applied in real-world screening, diagnosis, or intervention planning? This is particularly important for a journal like PLOS Mental Health, which values translational relevance.

4. Several important details (e.g., variable list, loadings, regression coefficients, fit indices) are only available in the supplementary materials. Please summarize key tables or figures in the main manuscript to improve transparency and reader accessibility

Reviewer #2: Very interesting and well-written. Only minor amendments are needed in my view.

L145 Can you distinguish consent from assent here if applicable

L147 Do you need approval for secondary analysis? If not, make it clearer that the original approval included secondary analyses.

L150 Classification into internalizing/externalizing as a % rather contradicts the focus on dimensional models. I would expect M (SD) of some z-score or similar here.

L157 Rewrite "Followingly," - not the right word.

L160 The data relies on parental reports, which will reduce reliability vs. teacher/peers and under-estimate some of the behaviours e.g. external, compulsions. This needs greater prominence in the discussion section.

L161 References need for the six questionnaires. Why didn't you use the PID-5 adolescent version?

L181 Assuming random stratification? Please state for clarity.

L230 How was pre/post-pubescent determined?

L278 Concerning variables that survived, was removal sequential (e.g. size of t value) or were decisions being made by humans?

L292 Do you mean "samples" rather than "sample" if two halves?

L307 Is "variance in the data" the right phrase? Variance in items?

L311 p-value = 1?

L327 "In the supplementary" - incomplete sentence

L442 The postpubescent data fitting better is unsurprising, because several constructs are more likely to be expressed more strongly here e.g. externalising, sexual tendencies in later years. This should be a discussion point.

L675 I was puzzled by this statement because surely any CA is a statistical summary of covariance of its items? This would apply to any model not just bifactor or g.

L696 Here and elsewhere, why the emphasis on internalising but not externalising symptoms - the PID-5 covers both. https://psycnet.apa.org/record/2019-70135-001

**Do you want your identity to be public for this peer review?** For information about this choice, including consent withdrawal, please see our Privacy Policy

Reviewer #1: **Yes: ** Samphoas Chien

Reviewer #2: **Yes: ** Gareth Hagger-Johnson

---

## [Decision Letter · Decision Letter 1]

12 Oct 2025

PMEN-D-25-00214R1

Toward Dimensional Psychiatry in Youth: A Data-Driven Analysis of Transdiagnostic Internalizing Symptoms in Childhood and Adolescence

PLOS Mental Health

Dear Dr. Engler,

Thank you for submitting your manuscript to PLOS Mental Health. After careful consideration, we feel that it has merit but does not fully meet PLOS Mental Health’s publication criteria as it currently stands. Therefore, we invite you to submit a revised version of the manuscript that addresses the points raised during the review process.

We look forward to receiving your revised manuscript.

Kind regards,

Juan Felipe Cardona, Ph.D.

Academic Editor

PLOS Mental Health

Journal Requirements:

Additional Editor Comments (if provided):

Reviewers' comments:

Reviewer's Responses to Questions

**Comments to the Author**

Reviewer #1: All comments have been addressed

Reviewer #2: All comments have been addressed

publication criteria?

Reviewer #1: Yes

Reviewer #2: Yes

3. Has the statistical analysis been performed appropriately and rigorously?

Reviewer #1: Yes

Reviewer #2: Yes

4. Have the authors made all data underlying the findings in their manuscript fully available (please refer to the Data Availability Statement at the start of the manuscript PDF file)?

Reviewer #1: Yes

Reviewer #2: Yes

5. Is the manuscript presented in an intelligible fashion and written in standard English?

Reviewer #1: Yes

Reviewer #2: Yes

Reviewer #1: I believe this article is well-written and sufficiently addresses the topic!

Reviewer #2: This is a good quality revision which addresses my previous concerns and suggestions. It is a nice paper and will open doors for the study team to do further papers using these constructs.

I have done a second fresh reading - apologies if I raise new things, I do not intend to generate extra work, but often read things differently on a resubmission. I have only minor suggestions below.

However, I think the results section needs cutting down at this point - there is far too much detail, and every decisions is described rather than the most important ones. It is good the analysis was diligently reported in full, and striking a balance is something we all struggle with, but it currently reads too much like a thesis rather than a journal article. Look at similar published papers and they are more concise. It is OK to move material to appendices, or just keep a record of less substantive decisions made in case asked by readers. Now it has been peer reviewed, we can trust the work has been done to a high standard, and lose some of the text. Perhaps the editor can make a judgement.

L132 Check journal requirements but I usually add commas to thousands e.g. 4,142 (here and throughout). Proofread for consistency throughout (and decide on approach in all tables, consistency needed).

L153 If known, I suggest added the % whose parents consented, but the participant did not assent and were excluded. It is good practice to report every step in which participants can be excluded (e.g. in flow diagram) and useful to understand if there were any differences here. Not essential, but if you have this information I would included it.

Table 1 - do not use 3 decimal places if 2 for other variables (I would suggest 1 throughout - nicer presentation and avoids pseudo-precision)

Table 1 - I strongly recommend never using dashes "-" in tables or when presenting confidence intervals, because these can be confused with minus signs. Use "to" word, or "," comma. You have used semicolon in Table 3 which is OK.

[P9 You may wish to explore, in future research, Mplus, which can do EFA/CFA with missing data quite well. It can also do "exploratory CFA" in one set of models. It is expensive but handles these sorts of scenarios nicely.]

L314 Be careful with "outcomes" which can imply causality. Even in longitudinal data, causality is not straightforward. Sometimes it is better to report the construct and the age (e.g. distress at age 18 or whatever) in case readers question bidirectional effects, reverse causation, residual confounding, time-varying confounding etc. In cross-sectional it is better not to say "outcomes" at all.

L339 Sequentially according to what? t value size?

L372 Again here and elsewhere, 3 decimal places is not necessary

L822 This is too apologetic - we all have to label constructs, in order to study and talk about them to further understanding. I would tone this down. It is the reliability/validity that will be important longer term, no one should get upset about the names particularly. We could spend our lives debating names - focus on their practical utility, predictive validity (the ultimate test).

L1244 There are 109 references - that is very high for a journal article (see above). You could lose anything up to a half I think.

Figure 1 - this should ideally separately show the entire eligible population, approached, recruited, consented, assented (if available). See CONSORT/STROBE guidelines. Clinicians are trained in critical appraisal (e.g. CASP) and this is one of the things they will scrutinize in detail - who are the population, who are the sample, are they different, in what ways? At every step.

Editor: I was not able to check the data and code availability at https://osf.io/br8wd/files/osfstorage. It was being upgraded at the time of writing ("We are verifying the upgrade to the OSF frontend codebase. Please refrain from using OSF until you see this message removed"). Perhaps you want to check availability before your decision.

**Do you want your identity to be public for this peer review?** For information about this choice, including consent withdrawal, please see our Privacy Policy

Reviewer #1: **Yes: ** Samphoas Chien

Reviewer #2: **Yes: ** Gareth Hagger-Johnson

---

## [Decision Letter · Decision Letter 2]

19 Nov 2025

Toward Dimensional Psychiatry in Youth: A Data-Driven Analysis of Transdiagnostic Internalizing Symptoms in Childhood and Adolescence

PMEN-D-25-00214R2

Dear Ms. Engler,

We are pleased to inform you that your manuscript 'Toward Dimensional Psychiatry in Youth: A Data-Driven Analysis of Transdiagnostic Internalizing Symptoms in Childhood and Adolescence' has been provisionally accepted for publication in PLOS Mental Health.

Best regards,

Juan Felipe Cardona, Ph.D.

Academic Editor

PLOS Mental Health

Reviewer Comments (if any, and for reference):

Reviewer's Responses to Questions

**Comments to the Author**

Reviewer #2: All comments have been addressed

publication criteria?

Reviewer #2: Yes

3. Has the statistical analysis been performed appropriately and rigorously?

Reviewer #2: Yes

4. Have the authors made all data underlying the findings in their manuscript fully available (please refer to the Data Availability Statement at the start of the manuscript PDF file)?

Reviewer #2: Yes

5. Is the manuscript presented in an intelligible fashion and written in standard English?

Reviewer #2: Yes

Reviewer #2: This is a good revision addressing all my concerns and comments.

I think it is ready for publication.

Very minor suggestions which I don't think need another round of reviews to fix (they are arguably typographical):

RMSEA: 0.0 [0.047; 0.048] (is this really 0.0 or is another significant figure needed)

"Clinical validity" (is this the right term, I haven't seen this used before. Could be confused with clinical significance, effectiveness in practice vs. efficacious etc.). Predictive validity in clinical settings?

Table 1 - not sure if journal will allow the male symbol (they can advise) but consider "Male (N, %") instead

**Do you want your identity to be public for this peer review?** For information about this choice, including consent withdrawal, please see our Privacy Policy

Reviewer #2: **Yes: ** Gareth Hagger-Johnson
